# A Minimally Invasive Approach for Preventing White Wine Protein Haze by Early Enzymatic Treatment

**DOI:** 10.3390/foods11152246

**Published:** 2022-07-28

**Authors:** Ilaria Benucci, Claudio Lombardelli, Massimo Muganu, Caterina Mazzocchi, Marco Esti

**Affiliations:** Department of Agriculture and Forest Sciences (DAFNE), Tuscia University, Via S. Camillo de Lellis snc, 01100 Viterbo, Italy; ilaria.be@unitus.it (I.B.); muganu@unitus.it (M.M.); caterina.mazzocchi@unitus.it (C.M.); esti@unitus.it (M.E.)

**Keywords:** white wine must, protein instability, haze-active proteins, microbial protease, haze prevention

## Abstract

Protein stability in bottled white wine is an essential organoleptic property considered by consumers. In this paper, the effectiveness of an early enzymatic treatment was investigated by adding a food-grade microbial protease at two different stages of winemaking: (i) at cold settling, for a short-term and low temperature (10 °C) action prior to alcoholic fermentation (AF); (ii) at yeast inoculum, for a long-lasting and medium temperature (18 °C) action during AF. The results reveal that protease sufficiently preserved its catalytic activity at both operational conditions: 10 °C (during cold settling) and 18 °C (during AF). Furthermore, protease addition (dosage 50–150 μL/L) raised the alcoholic fermentation rate. The treatment at yeast inoculum (dosage 50 μL/L) had a remarkable effect in preventing haze formation, as revealed by its impact on protein instability and haze-active proteins. This minimally invasive, time and resource-saving enzymatic treatment, integrated into the winemaking process, could produce stable white wine without affecting color quality and phenol content.

## 1. Introduction

The formation of protein haze in bottled white wine is affected by several known factors, including the presence of phenolic compounds, polysaccharides, sulfate and metal ions, ionic strength as well as by the pH and the wine storage temperature. However, other important factors remain unidentified [1]. Although the exact mechanism of protein haze formation is not very clear, it may be assumed that it is mainly associated with hydrophobic protein–phenolic interactions [2]. Furthermore, it has been observed that pathogenesis-related (PR) proteins (e.g., thaumatin-like proteins TLP and chitinase CH) derived from grapes are mainly responsible for haze formation in bottled white wine [3]. Van Sluyter et al. [4] have proposed a revised mechanism of protein haze formation involving three different stages. Such mechanism is based on the discovery that TLP and CH have different unfolding temperatures and unfolding/aggregation behavior, as well as the interaction of non-proteaceous components (salts, sulphate and phenolics) in the late phase.

The most used method to treat white wine instability is the batch addition of bentonite at the end of wine ageing before bottling. This required treatment, in addition to being an additional time- and resource-consuming step, is associated with negative environmental impact (disposal of spent materials), product loss (3–10%) and quality degradation of wine at the end of its maturation [1,3], estimated to be about USD 1 billion per year [5]. To overcome these criticisms, several approaches have been studied to solve the question of protein instability in white wine, all of them suggesting their application on finished wine. Among the innovative methods based on a physical approach, it is possible to include the use of ultrasound [6]. In recent years, countless works have been included in the research of stabilization methods, always as an additional process step based on the use of porous adsorbent materials [5]: acrylic acid plasma-coated magnetic nanoparticles [7], zeolites [8], grape seeds powder [9] and zirconium oxide. Research is needed to determine if the components of these adsorbent materials (e.g., magnetic nanoparticles and acrylic acid) are released into the wine or if the compounds in the wash may permeate the material and leach into the wine. Furthermore, improved regeneration methods should be associated with the use of porous adsorbents in order to lower waste and water needed to clean the materials [5]. Additive stabilizing approaches to prevent haze formation have been suggested and consist in using agents of animal/vegetable origins, such as mannoproteins (50% potential instability reduction [10]), carrageenan/pectin (75–90% protein reduction [11]) and chitosan (14% protein reduction [12]). Most of these techniques diminish the protein content and do not affect the sensory characteristics of the wine, but some issues must be considered (e.g., alteration of the metal ion content and generation of precipitates/residues that could affect other stages of wine production, such as filtration) [5]. Unstable protein fractions in white wine have been the target of studies applying proteolytic enzymes instead of traditional bentonite fining to avoid protein haziness [13]. Proteases must be able to work under restricted wine conditions of pH, alcohol content and temperature. The use of a fungal endoprotease (aspergilloglutamic protease) following flash pasteurization was tested in must and white wine by Marangon et al. [14]. Plant-based proteases have also been extensively studied [1]: papain from papaya [15,16] and bromelain from pineapple [17] have been tested concerning their ability in the degradation of heat-unstable white wine proteins. Bromelain exhibited effectiveness in the degradation of wine proteins (approximately 70%) in model wine as well as in real white wine, also when immobilized on chitosan beads and applied in a laboratory-scale stirred reactor [18,19].

The novelty and significant contribution of this study consists of providing an enzymatic treatment to be applied in must of white grape, since it naturally already contains the targeted proteins extracted from grape (TLP and CH) and its content of endogenous/exogenous inhibitors is still limited. A microbial protease was therefore applied in must of white grape at two different stages at the beginning of winemaking: (i) at cold settling, for a short-term and low temperature (10 °C) action prior to alcoholic fermentation (AF); (ii) at yeast inoculum, for a long-lasting and medium temperature (18 °C) action during AF. Such early protein stabilization, performed on grape must at the initial stage of winemaking and integrated into the production process (not as an additional step), is deemed to be minimally invasive given that wine is still evolving.

## 2. Materials and Methods

### 2.1. Enzymes, Chemicals and Must

A commercial food-grade protease from *Aspergillus niger* was kindly provided by IMCD Italia SpA (Milan, Italy); the chromogenic substrate (Z-Gly-Pro-pNA) was purchased from Bachem (Bubendorf, Switzerland) and the active dry yeast (*Saccharomyces cerevisiae*, Zymaflore^®^) used for the experimental fermentation was purchased by Laffort (Tortona, Italy). All other reagents were from Merck Group (Milan, Italy). The must from *Vitis vinifera* L. cv Riesling italic grape was kindly supplied by Castello delle Regine (Terni, Italy) and its main oenological parameters were determined according to European official methods of analysis [20].

### 2.2. Enzymatic Activity Assay and Kinetic Study

Protease activity was spectrophotometrically assessed (λ: 410 nm, Shimadzu UV 2450, Milan, Italy) at two different temperatures representing the operational conditions (10 °C (cold settling prior to AF) and 18 °C (AF)). The assay was performed in model wine must (tartaric buffer 0.3 M, pH 3.2) to which was added the synthetic chromogenic substrate Z-Gly-Pro-pNA, prepared as described by Lopez et al. [21]. For the kinetic characterization, the substrate concentration was varied from 0.03 to 1.0 mM and the kinetic parameters V_max_ (maximum initial velocity) and K_M_ (Michaelis–Menten constant, corresponding to the substrate concentration when the initial velocity is one-half of the V_max_) were estimated according to the Michaelis–Menten equation by a non-linear regression (GraphPad Prism 5.01, GraphPad Software, Inc., La Jolla, CA, USA). k_cat_ (turnover number, indicating the number of moles of substrate converted into the product per number of moles of catalyst per minute) and K_a_ (affinity constant, k_cat_/K_M_, suggesting the affinity of the enzyme toward the substrate) were calculated as reported by Benucci et al. [18].

### 2.3. Enzymatic Treatment and Laboratory-Scale Fermentation Tests

Protease was applied in the range of 2–150 μL/L at two different winemaking phases: (i) at cold settling, for a short-term and low temperature (10 °C) action prior to alcoholic fermentation (AF) (treatment I); (ii) at yeast inoculum, for a long-lasting and medium temperature (18 °C, temperature usually used in white wine AF) action during AF (treatment II). The effectiveness of proteolytic dosage was evaluated at racking after 24 h settling (treatment I) and at middle/end of AF (treatment II). A detailed description of the treatments applied is summarized in Table 1. Treatment I was conducted dividing the must into 200 mL aliquots inside graduated cylinders, adding the different doses of proteolytic enzyme (2, 5, 10, 30, 50, 70, 100 and 150 μL/L). Then, the samples were subjected to cold settling prior to AF (10 °C, overnight) after also adding the pectinolytic preparation (3 mL/hL). Samples were compared with the grape juice settled with only pectinolytic enzyme (Ctrl). Treatment II was conducted on clear must (after cold settling with pectinolytic enzyme) adding protease (2, 5, 10, 30, 50, 70, 100 and 150 μL/L) at yeast inoculum and compared with the clear must fermented without any enzyme addition (Ctrl). The laboratory-scale fermentation tests were carried out as reported by Benucci et al. [22] using commercial active dry yeast (*Saccharomyces cerevisiae* Zymaflore^®^ 20 g/hL). Treatments were performed in triplicate. AF was performed at 18 °C for about 16 days and weight loss was used as parameter for monitoring the evolution of fermentation process. The dynamics of weight loss were fitted by means of a sigmoid or altered Gompertz decay function as described by Benucci et al. [23]. The fermentation rate (K) and half-time (M) were estimated by a non-linear regression procedure (GraphPad Prism 5.0, GraphPad software, Inc., La Jolla, CA, USA).

### 2.4. Heat Stability and Haze-Active (HA) Protein Determination

The haze potential (ΔNTU) of white wine must was calculated as the difference between sample turbidity (HD 25.2 turbidimeter, Delta OHM, Padua, Italy) after heat test (80 °C for 6 h and 4 °C for 16 h [24]) and its initial turbidity [18]. HA proteins were detected as described by Siebert et al. [25] and Benucci et al. [26] using tannic acid (1 mg/mL) at 25 °C after 30 min. The haze difference between the sample with and without added tannic acid was reported as the HA proteins.

### 2.5. Total Protein Content and Electrophoretic Separation (SDS-PAGE)

Total protein content of the samples was determined using the potassium dodecyl sulphate method [24] and protein quantification was carried out using the bicinchoninic acid (BCA) protein assay kit (Sigma-Aldrich, Saint Louis, MO, USA) [18]. Electrophoretic separation (SDS-PAGE) was performed using a vertical electrophoresis apparatus (Mini-Protean Tetra cell, Bio-Rad, Richmond, CA, USA) and standard molecular weight (Precision Plus Protein Standards, Kaleidoscope, Bio-Rad, Richmond, CA, USA) [18].

### 2.6. Effect of Enzymatic Treatment on Chromatic Characteristics and Phenolics

The concentration of total phenols was determined as reported by Becchetti et al. [27]. In brief, 10 mL of Na_2_CO_3_ (20% *w*/*v*) and 5 mL of Folin–Ciocalteau reagent were added to 1 mL of wine must and lead to 100 mL with distilled water. After 30 min, the absorbance at 700 nm was registered and results were expressed as gallic acid equivalent (mg/L). CIELAB parameters (hue (h), chroma (C*) and total color differences (ΔE*)) were also assessed using a CR-5 colorimeter (Konica Minolta, Tokyo, Japan) by a D65 illumining and CIELAB uniform color space [28]. ΔE* was calculated referring to Ctrl sample.

### 2.7. Statistical Analysis

Data provided are the mean of triplicate analytical determinations. One-way ANOVA (*p* < 0.01) and Tukey’s HSD test (*p* < 0.05) were applied to determine any statistical differences among samples (EXCEL^®^ Add-in macro DSAASTAT program) [29]. The 95% confidence intervals presented in Tables were estimated by using GraphPad Prism 5.0.

## 3. Results and Discussion

### 3.1. Enzyme Kinetic Study

Before testing the effectiveness of protease in preventing wine haze, its kinetic behavior was studied in model wine must with the addition of a synthetic chromogenic substrate at two different temperatures representing the operational conditions: 10 °C (cold settling prior to AF) and 18 °C (during AF). As expected, the protease showed the hyperbolic behavior of the Michaelis–Menten equation (Figure 1) and the corresponding kinetic parameters are reported in Table 2.

Although the similar values of V_max_, the other parameters (K_M_, k_cat_ and K_a_) proved a better catalytic efficiency at 18 °C with respect to 10 °C. However, also at low temperature (10 °C), protease sufficiently preserved its catalytic activity (Table 2). These results are in line with Kang et al. [30] who found a 10% relative activity loss for prolyl endopeptidase purified from A. oryzae in the temperature range 15–30 °C using the same substrate (Z-Gly-Pro-pNA) in phosphate buffer (pH 5.0).

### 3.2. Alcoholic Fermentation Kinetic

The kinetic of sugar consumption in wine must (24° Brix corresponding to 20.7° Babo; pH = 3.47 ± 0.01; total acidity = 5.24 ± 0.01 g_tartaric acid_/L; potential alcohol content = 14.0% *v*/*v*) suggests remarkable differences among samples (Figure 2). The lowest fermentation rate (K) was observed for Ctrl, whereas in all the treated samples it increased as the enzyme amount was raised, especially for enzyme dosages above 50 μL/L.

Considering K and M values (Table 3), it is possible to notice an increase in the K values and a corresponding decrease in M values at the highest protease dosage (from 50 to 150 μL/L). This trend could be explained considering that the proteolytic enzyme treatment may have affected yeast vitality and consequently the velocity of sugar consumption during alcoholic fermentation. This phenomenon may be ascribed to the enzymatic hydrolysis of the proteins in wine must and the consequent increase in the peptides/amino acids available as an assimilable nitrogen source for yeast metabolism. Lei et al. [31] found that protease (Flavorzyme) supplementation to high gravity beer was beneficial for the success of alcoholic fermentation, both in terms of wort fermentability and higher ethanol yield. Mathias et al. [32] demonstrated that the protease addition promoted the increase in total nitrogen and amino acid content in the sweet wort and a higher fermentation efficiency in brewing.

### 3.3. Effect of Enzymatic Treatment on Protein Instability

In order to prevent haze formation in white wine, a microbial protease was applied at two different early winemaking phases: (i) at cold settling and (ii) at yeast inoculum.

Concerning treatment I, a significant reduction in protein instability was found adding protease (Figure 3a) at dosages above 10 μL/L. The lowest ΔNTU (5.1) was observed at 50 μL/L, whereas no further stabilizing effect was revealed at highest dosages. The cold settling in presence of the pectinolytic enzyme alone (Ctrl) lowered the protein instability (ΔNTU = 15) as compared to the turbid pressed grape juice (ΔNTU = 21.3, data not shown). Similar results were observed for the treatment II (evaluated at the middle/end of AF), with the lowest ΔNTU values (5.9 and 4.1, respectively) at 50 μL/L, corresponding to a 75–83% instability removal (Figure 3b). No further ΔNTU reduction was found at increasing proteolytic enzyme dosage (Figure 3b). Several studies have investigated the effectiveness of different proteolytic enzymes in free or immobilized form as an alternative treatment to bentonite fining for white wine protein stabilization. Benucci et al. [17] tested the feasibility of stem bromelain, free or immobilized on chitosan support, to reduce white wine protein haze potential (approximately 70%). Marangon et al. [14] demonstrated that the simple addition of AGP (a mixture of Aspergillopepsins I and II) during fermentation yielded a protein hydrolysis of about 20%.

Similar to that observed for protein instability, the decrease in HA proteins appeared remarkable adding protease (Figure 4). Irrespective of the oenological phase, the proteolytic enzyme had a significant effect in breaking down HA proteins (Figure 4a,b) and the most effective dosage resulted as 50 μL/L (0.6 NTU). The efficacy of protease and pectinase in beverages has been highlighted by the literature. Both enzymes significantly decreased the amount of HA proteins (−75%) in pomegranate juice treated in fluidized bed reactor [26].

In both enzymatic processes, a significant difference in total protein appeared between the Ctrl and treated samples (−20% and −32% for treatment I and II, respectively), whereas no remarkable discrepancy among dosages were revealed (Figure 5a,b). Total protein amount decreased to a greater extent when protease was added at yeast inoculum and left in wine must during all the AF duration (treatment II). Therefore, it is possible to hypothesize that this protein depletion may be attributed to the combined action exerted by the proteolytic enzyme throughout a long-lasting contact time at higher temperature, as well as by fermenting yeast metabolism and co-precipitation phenomena. It may be presumed that, at cold settling (treatment I), the most unfavorable conditions both in terms of temperature (10 °C compared to 18 °C of AF) and time (24 h as compared to 16 days) were detrimental for enzyme activity.

Before and after the enzymatic treatment, the wine samples were subjected to the SDS-page and the relative electrophoretic profiles, obtained under denaturing conditions, are depicted in Figure 6. The reduction in the bands between 20–25 kDa, corresponding to CH (25 kDa) and TLP (22 kDa) [14,33], after enzymatic treatment proved its effectiveness in both winemaking phases (Figure 6).

### 3.4. Effect of Enzymatic Treatment on Chromatic Characteristics and Phenolics

Table 4 and Table 5 show the colorimetric parameters L*, hue (h*), chroma (C*) and total color difference (ΔE) (obtained on the basis of the CIELAB space coordinates (L*, a*, b*)). Irrespective of the oenological phase of enzyme addition, L* showed no significant differences (Table 4 and Table 5), while h* values were in the range of 81.7–85.9, confirming that the main shade of wine samples was in the yellow portion. No relevant differences were revealed either with respect to Ctrl or among the enzymatically treated samples. For both treatments, C* parameter (Table 4 and Table 5) was similar among the samples and Ctrl. All values indicated that the color of the samples fallen into the “pale-yellow” color category [34]. ΔE parameter in all treated samples was around 3, thus suggesting the lack of a remarkable difference in color between samples and Ctrl (Table 4 and Table 5). As reported by Lukić et al. [34], to have an appreciable difference in color between white wine samples, the ΔE must be greater than 3.5.

The phenolic content of wine must produced by Riesling Italic grapes (Figure 7) resulted in the range of 200–280 mg/L, in line with what is reported in the literature [35]. Concerning treatment I (Figure 7a), no appreciable differences were observed among samples, whereas a slight decrease in phenol content was revealed as compared to the Ctrl following treatment II in all samples at the end of AF (Figure 7b). It is reasonable to assume the phenomenon underlying this decline is the precipitation of plant residues, tartaric salts, as well as the sorption of grape high molecular weight phenols by yeast cell walls [36], also related to ΔE changes.

Overall, these results suggest that the novel protein stabilization treatment did not significantly affect the phenolic composition of wine, as already demonstrated by applying other biocatalysts [19].

## 4. Conclusions

According to the results of this study, the early protein stabilization treatment, performed at the initial stages of winemaking, was useful in lowering the protein instability and the amount of HA proteins in wine must from white grape, without affecting color quality and phenol content. Such early treatment could contribute to the protein stabilization naturally occurring later throughout the wine ageing on lees. In summary, the addition of microbial protease at yeast inoculum (performed at 50 μL/L dosage) appeared to be the most useful in preventing protein haze in comparison with the enzymatic treatment at cold settling, probably due to the most favorable conditions for protease activity both in terms of temperature (18 °C of AF as compared to 10 °C of settling) and time (16 days as compared to 24 h). Despite the fact there are more sustainable approaches for winemaking, there should also be further investigation. This minimally invasive enzymatic treatment integrated into the production process could represent a valuable alternative to conventional bentonite fining as well as to the most recently available stabilization methods, always intended as an additional process step.

## Figures and Tables

**Figure 1 foods-11-02246-f001:**
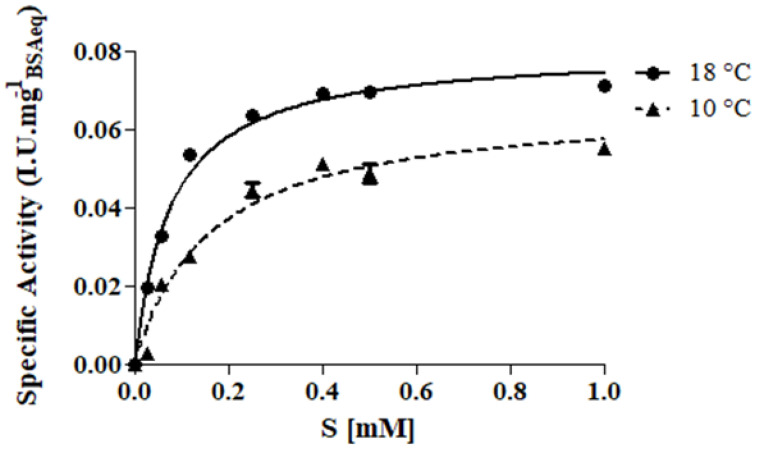
Kinetic curves of protease tested at two different temperatures: 10 °C (▲, cold settling prior to AF) and 18 °C (●, AF). Assay was performed in model wine-must (tartaric buffer 0.3 M, pH 3.2) added with the synthetic chromogenic substrate Z-Gly-Pro-pNA (0–1 mM).

**Figure 2 foods-11-02246-f002:**
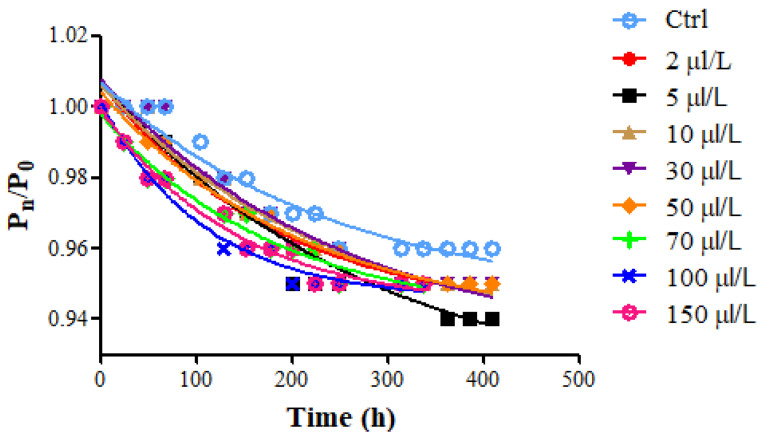
Time course of weight loss (ratio between the current and initial weight (P_n_/P_0_) against time (t)) representing sugar consumption during the alcoholic fermentation (18 °C) of Riesling Italic must (treatment II, at yeast inoculum) at different protease dosages (2–150 μL/L).

**Figure 3 foods-11-02246-f003:**
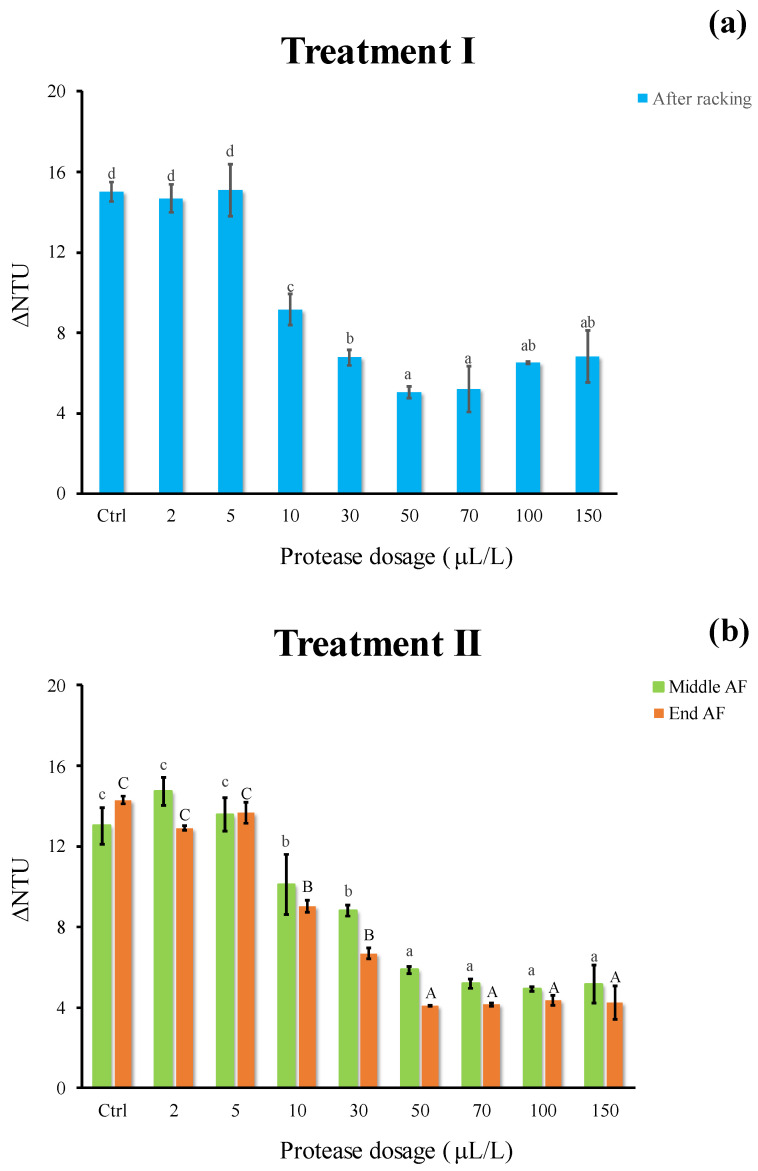
Haze potential of white must wine (ΔNTU) subjected to enzymatic addition (protease in the range of 2–150 μL/L) after (**a**) treatment I (cold settling prior to AF) and (**b**) treatment II (at yeast inoculum). Ctrl is clarified must with the only pectinolytic enzyme. For each sample, means with different roman letters are significantly different (*p* ≤ 0.05).

**Figure 4 foods-11-02246-f004:**
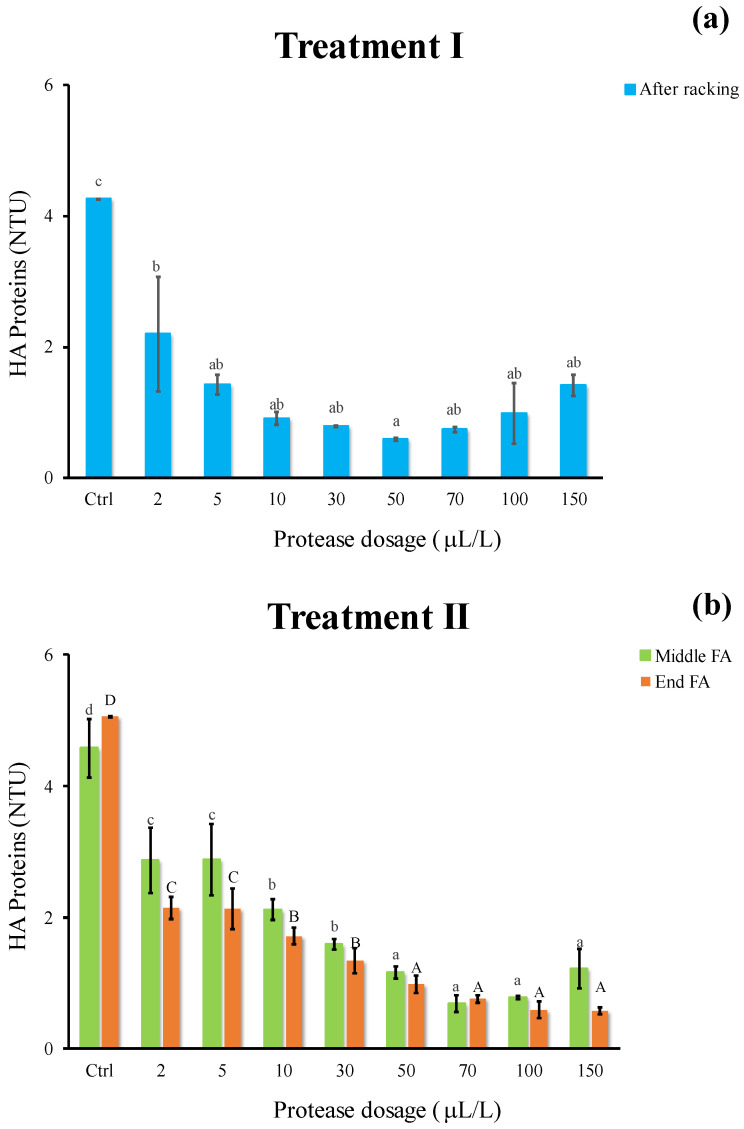
Haze active (HA) proteins of white must wine (NTU) subjected to enzymatic addition (protease in the range of 2–150 μL/L) after (**a**) treatment I (cold settling prior to AF) and (**b**) treatment II (at yeast inoculum). Ctrl is clarified must with the only pectinolytic enzyme. For each sample, means with different roman letters are significantly different (*p* ≤ 0.05).

**Figure 5 foods-11-02246-f005:**
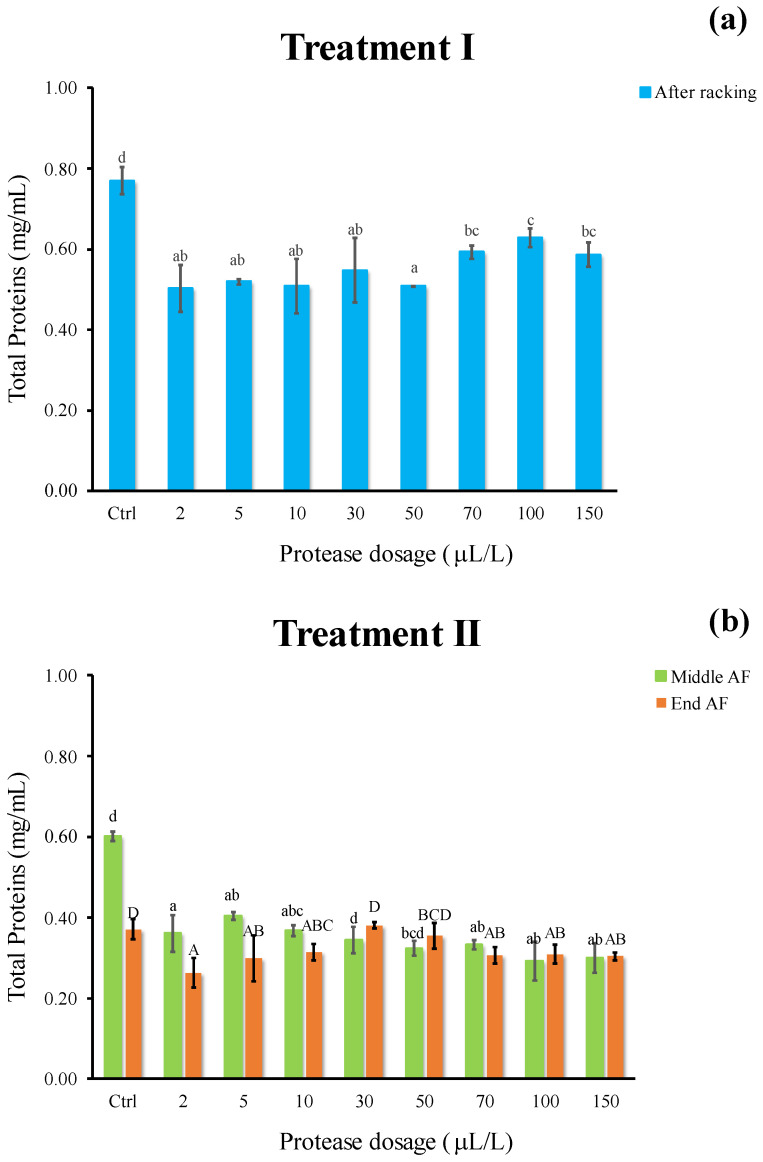
Total proteins (mg/mL) of white must wine subjected to enzymatic addition (protease in the range of 2–150 μL/L) after (**a**) Treatment I (cold settling prior to AF) and (**b**) Treatment II (at yeast inoculum). Ctrl is clarified must with the only pectinolytic enzyme. For each sample, means with different roman letters are significantly different (*p* ≤ 0.05).

**Figure 6 foods-11-02246-f006:**
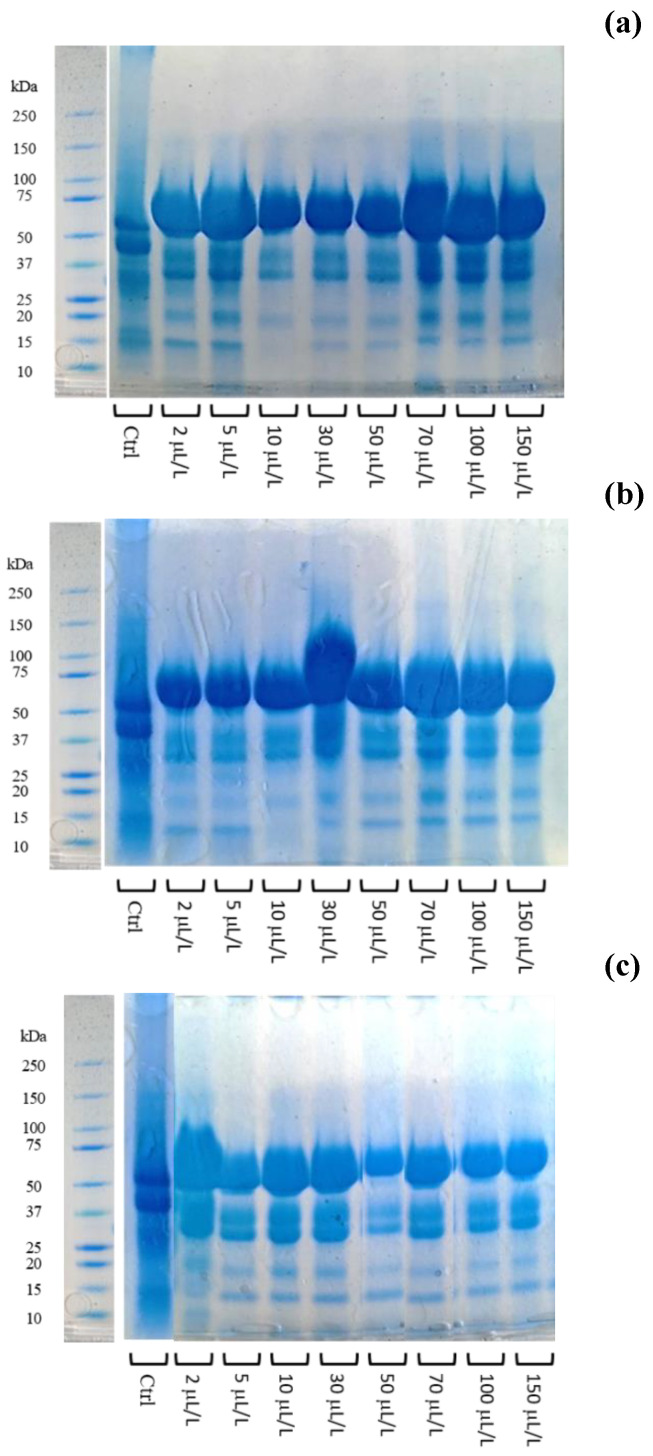
Electrophoretic profile under denaturing conditions of white must-wine subjected to enzymatic addition (protease in the range of 2–150 μL/L) after (**a**) treatment I (cold settling prior to AF, evaluated at racking after 24 h settling), (**b**) treatment II (at yeast inoculum, evaluated at middle AF) and (**c**) treatment II (at yeast inoculum, evaluated at the end AF). Ctrl is clarified must with the only pectinolytic enzyme.

**Figure 7 foods-11-02246-f007:**
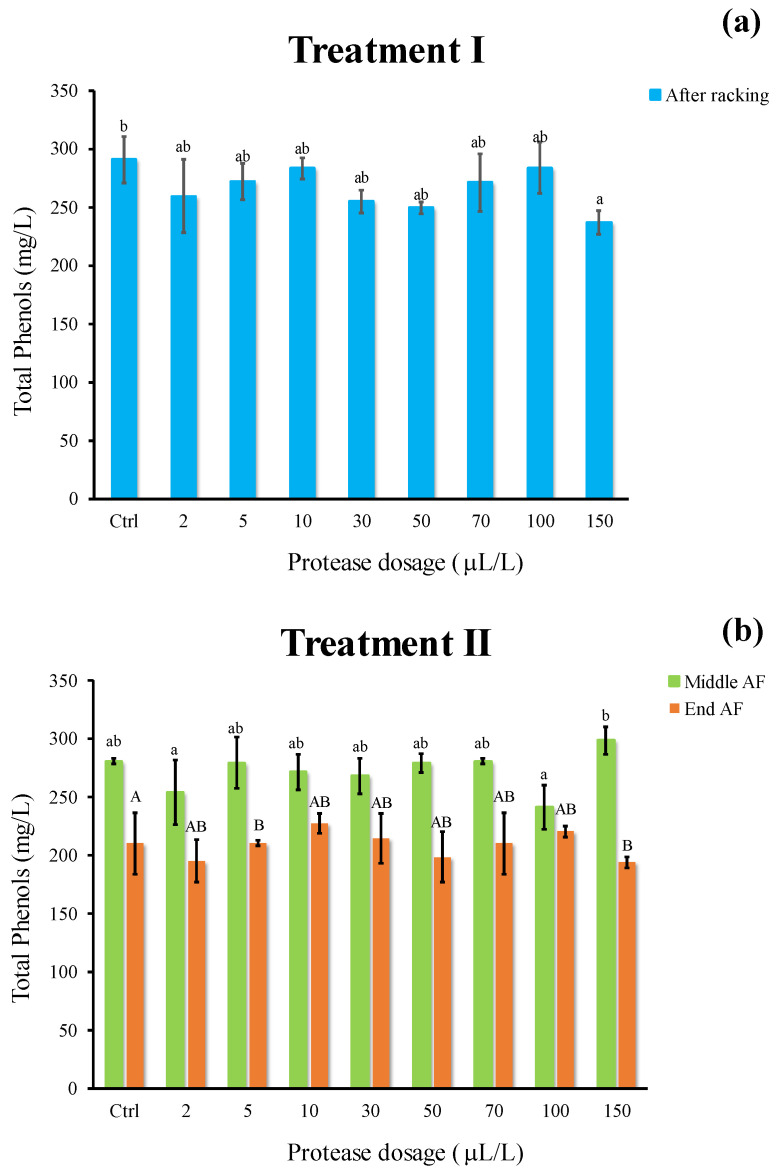
Total phenols (mg/L) of white must wine subjected to enzymatic addition (protease in the range of 2–150 μL/L) after (**a**) treatment I (cold settling prior to AF) and (**b**) treatment II (at yeast inoculum). Ctrl is clarified must with the only pectinolytic enzyme. For each sample, means with different roman letters are significantly different (*p* ≤ 0.05).

**Table 1 foods-11-02246-t001:** Detailed frame of the applied treatments.

	Description	Duration	Temperature	Reference Sample (Ctrl)	Analysis Time
**Treatment I**	Addition of protease before cold settling prior to AF	24 h	10 °C	Grape juice settled with only pectinolytic enzyme	After racking
**Treatment II**	Addition of protease at yeast inoculum	Overall AF duration (16 days)	18 °C	Clear must after cold settling (with pectinolytic enzyme) fermented without any enzyme addition	Middle AF End AF

**Table 2 foods-11-02246-t002:** Kinetic parameters of protease tested at two different temperatures: 10 °C (cold settling prior to AF) and 18 °C (AF). Assay was performed in model wine must (tartaric buffer 0.3 M, pH 3.2) added with the synthetic chromogenic substrate Z-Gly-Pro-pNA (0–1 mM).

	T = 10 °C	T = 18 °C
**V_max_ (I.U.mg^−1^_BSAeq_)**	0.066 ± 0.003	0.080 ± 0.001
**K_M_ (mM)**	0.160 ± 0.015	0.075 ± 0.005
**k_cat_ (min^−1^)**	8.38 × 10^2^ ± 0.003	1.02 × 10^3^ ± 0.001
**Ka (min^−1^ mM^−1^)**	5.24 × 10^3^ ± 49.12	1.35 × 10^4^ ± 910.7
**R^2^**	0.96	0.99

**Table 3 foods-11-02246-t003:** Parameters obtained by fitting the altered Gompertz equation to the experimental data of weight loss. K: alcoholic fermentation rate; M: half-time of weight loss.

Protease Dosage (μL/L)	K (g/h)	M (1/h)
**Ctrl**	0.0041 (0.0034–0.0043)	169.9 (160.8–179.0)
**2**	0.0038 (0.0031–0.0042)	178.6 (169.0–188.2)
**5**	0.0036 (0.0032–0.0041)	192.9 (169.5–203.3)
**10**	0.0041 (0.0034–0.0043)	168.8 (159.7–179.6)
**30**	0.0039 (0.0034–0.0041)	176.4 (166.9–185.9)
**50**	0.0047 (0.0044–0.0049)	146.7 (144.1–149.3)
**70**	0.0053 (0.0049–0.0058)	129.9 (127.8–132.3)
**100**	0.0091 (0.0087–0.0093)	76.24 (74.9–79.6)
**150**	0.0090 (0.0086–0.0094)	76.20 (74.6–79.8)

Reported values are mean ± 95% confidence interval of triplicate measurements.

**Table 4 foods-11-02246-t004:** Visual color attributes (L*, hue (h*), chroma value (C*) and total color difference (ΔE)) of white must wine subjected to enzymatic addition (protease in the range of 2–150 μL/L) after treatment I (cold settling prior to AF). Ctrl is clarified must with the only pectinolytic enzyme.

Protease Dosage (µL/L)	L*	h*	C*	ΔE
**Ctrl**	95.5 (95.1–95.9)	81.1 (79.6–82.5)	9.3 (8.9–10.8)	-
**2**	95.3 (94.9–95.7)	81.2 (79.7–82.6)	9.0 (8.8–9.2)	0.27 (0.29–0.31)
**5**	95.3 (94.7–95.9)	81.4 (79.9–82.8)	8.9 (8.7–9.1)	0.30 (0.28–0.31)
**10**	95.3 (94.6–95.8)	81.7 (80.2–83.1)	8.9 (8.8–9.3)	0.40 (0.39–0.41)
**30**	96.4 (96.0–96.8)	81.7 (80.3–83.3)	8.9 (8.6–9.1)	0.91 (0.87–0.93)
**50**	95.4 (95.0–95.8)	81.7 (80.1–82.9)	9.0 (8.8–9.2)	0.93 (0.91–0.95)
**70**	96.8 (96.4–97.2)	82.2 (80.7–83.6)	10.6 (10.4–10.8)	1.2 (1.0–1.4)
**100**	96.4 (95.9–96.9)	82.4 (80.9–83.9)	10.6 (10.3–10.7)	1.0 (0.8–1.3)
**150**	96.5 (96.1–96.9)	81.9 (80.4–83.4)	10.4 (10.2–10.6)	1.3 (1.0–1.4)

Reported values are mean ± 95% confidence interval of triplicate measurements. “-”: not detectable.

**Table 5 foods-11-02246-t005:** Visual color attributes (L*, hue (h*), chroma value (C*) and total color difference (ΔE)) of white must wine subjected to enzymatic addition (protease in the range of 2–150 μL/L) after treatment II (at yeast inoculum, evaluated at middle AF and at the end of AF). Ctrl is clarified must with the only pectinolytic enzyme.

	L*		h*		C*		ΔE	
Protease Dosage (µL/L)	Middle AF	End AF	Middle AF	End AF	Middle AF	End AF	Middle AF	End AF
**Ctrl**	98.5 (98.1–98.9)	95.4 (95.0–95.8)	83.6 (82.1–85.1)	84.8 (83.3–86.3)	8.9 (8.7–10.1)	8.5 (7.9–9.3)	-	-
**2**	98.1 (97.7–98.5)	95.2 (94.8–95.6)	81.8 (80.3–83.3)	81.8 (80.1–83.5)	8.8 (8.6–9.0)	8.2 (8.0–8.4)	2.8 (2.6–2.9)	3.4 (3.2–3.6)
**5**	97.4 (97.0–97.8)	95.2 (94.7–95.7)	82.3 (80.8–83.8)	82.1 (80.6–83.6)	9.0 (8.8–9.2)	8.1 (7.9–8.3)	2.8 (2.5–2.9)	3.4 (3.2–3.5)
**10**	97.8 (97.4–98.2)	95.3 (94.9–95.7)	85.9 (84.4–87.4)	81.3 (79.9–82.8)	8.9 (8.5–9.0)	7.9 (7.7–8.2)	2.7 (2.5–2.8)	3.3 (3.1–3.5)
**30**	98.1 (97.6–98.6)	95.2 (94.6–95.7)	85.9 (84.5–87.7)	82.2 (80.7–83.7)	9.1 (8.9–9.3)	8.3 (8.0–8.6)	2.6 (2.4–2.7)	3.3 (3.2–3.5)
**50**	97.9 (97.5–98.3)	95.2 (94.7–95.6)	85.1 (83.6–86.6)	81.8 (80.1–83.6)	9.5 (9.3–9.7)	8.3 (8.1–8.5)	2.7 (2.6–2.9)	3.2 (3.0–3.4)
**70**	97.0 (96.6–97.4)	94.9 (94.5–95.3)	84.4 (82.9–85.9)	83.2 (81.7–84.7)	10.1 (9.9–10.3)	9.1 (8.8–9.2)	2.5 (2.4–2.7)	3.2 (3.0–3.5)
**100**	97.2 (96.8–97.6)	94.9 (94.6–95.4)	84.6 (83.1–86.1)	83.3 (81.8–84.8)	9.9 (9.6–10.5)	9.2 (8.9–9.4)	2.6 (2.3–2.7)	3.3 (3.2–3.6)
**150**	97.8 (97.4–98.2)	95.0 (94.6–95.4)	83.8 (82.3–85.3)	83.1 (81.6–84.6)	9.9 (9.7–10.1)	8.9 (8.6–9.1)	2.7 (2.5–2.8)	3.4 (3.1–3.6)

Reported values are mean ± 95% confidence interval of triplicate measurements. “-”: not detectable.

## Data Availability

The data presented in this study are available upon request from the corresponding author.

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
