# Peer review of "A Minimally Invasive Approach for Preventing White Wine Protein Haze by Early Enzymatic Treatment"

_foods, 2022, doi:10.3390/foods11152246_

Round 1
Reviewer 1 Report
Manuscript ID: foods-1813247
Title: A minimally invasive approach for preventing white wine protein haze by early enzymatic treatment
The topic addressed in this article is worthy of investigation. However, the approach of the study does not provide clear results with respect to the objective described. The authors cannot conclude that the observed effects are due to the addition of microbial protease, since pectinolytic preparation are also added to one of the treatments proposed for the study.
Page 3, Lines15-106: The difference between control and untreated must is not clear. Normally the word control is used to refer to the sample that has not been subjected to any of the treatments under test. All samples tested must be compared to the same reference for the results to be interpretable.
Page 3, Line 106: An experimental design must reflect all the tests to be carried out based on the variables considered. In table 1, only the two treatments to which the samples are subjected are observed, but do not consider other variables such as the presence or absence of pectolytic preparation or the amount of microbial protease added.
The authors do not detail the replicates that are made for each test nor in the analytical determinations.
Page 5, Line 161: The authors do not define the kinetic parameters neither in this section nor in materials and methods.
Page 5, Lines174-176: There are two variables: pectolytic preparation and amount of microbial protease. What is really the effect observed? Are you sure that the effect is due to the presence of microbial protease?
Figure 2: Pn/Po is not described or defined in any section of the work, nor in the figure caption.
Page 7, Lines 204-207. The observed effect is mainly due to the presence of pectinolytic enzymes, right?
Figure 3. What exactly is ΔNTU and how has it been calculated for each sample in both treatments? It does not seem that the reference is the same and, therefore, the results would not be comparable.
Ctrl is defined differently in the different figures. Is each case a different sample?
The figure 4a confirms the idea that the observed effect is mainly due to the presence of pectolytic enzymes and these are not present in all the samples, so the conclusions obtained in this work should be questioned.
Page 11, Lines 272-274. Two different effects are then compared since the two treatments are not carried out under the same conditions in relation to the presence/absence of pectolytic enzymes. Two variables should be considered for the study to be complete and the results obtained objective.
Figure 7a: Indicators of significant differences do not appear
Author Response
The topic addressed in this article is worthy of investigation. However, the approach of the study does not provide clear results with respect to the objective described. The authors cannot conclude that the observed effects are due to the addition of microbial protease, since pectinolytic preparation are also added to one of the treatments proposed for the study.
R: We would like to thank the Reviewer for her/his observation, giving us the chance for better describe the methodology applied. As now clearly indicated in the text (section 2.3), pectinolytic preparation has been added (as usual settling adjuvant treatment before alcoholic fermentation in white winemaking) to both the treatments proposed and deemed as reference sample (please see the elucidation reported below).
- Page 3, Lines15-106: The difference between control and untreated must is not clear. Normally the word control is used to refer to the sample that has not been subjected to any of the treatments under test. All samples tested must be compared to the same reference for the results to be interpretable.
R: We apologize for the lack of clarity. The untreated must (previously indicated TQ was the turbid grape juice immediately after pressing and cold-settled without any enzyme addition). The control (indicated as Ctrl) was the cold-settled clarified must added with the only pectinolytic enzyme.
We agree with Reviewer 1 about the need for more clarity, and for this reason in both treatments (I and II) all samples treated with protease at different dosage (besides the usual settling adjuvant treatment based on pectinolytic enzyme) are now compared to the same reference (Ctrl, clarified must cold-settled with the only pectinolytic enzyme addition).
- Page 3, Line 106: An experimental design must reflect all the tests to be carried out based on the variables considered. In table 1, only the two treatments to which the samples are subjected are observed, but do not consider other variables such as the presence or absence of pectolytic preparation or the amount of microbial protease added.
The authors do not detail the replicates that are made for each test nor in the analytical determinations.
R: We would like to thank the Reviewer for her/his observation. We erroneously indicated Table 1 as an experimental design. Table 1 has been included in the text for better describe/reassume the details of treatments (I and II). Its caption has been rewritten to avoid confusion.
As now indicated, treatments were performed in triplicate (section 2.3) and all data are the mean of triplicate analytical determinations (section 2.7).
- Page 5, Line 161: The authors do not define the kinetic parameters neither in this section nor in materials and methods.
R: We apologies for the omission. A clear definition of the kinetic parameters has been included in section 2.2.
- Page 5, Lines174-176: There are two variables: pectolytic preparation and amount of microbial protease. What is really the effect observed? Are you sure that the effect is due to the presence of microbial protease?
R: Sorry for the lack of clarity. The only variable is the amount of microbial protease added at two different winemaking phases. As now explained in the text (section 2.3) the pectinolytic preparation was applied in both treatments (I and II). Thus, the effect observed may be uniquely ascribable to the presence of microbial protease added at different dosage.
-Figure 2: Pn/Po is not described or defined in any section of the work, nor in the figure caption.
R: We apologize for the omission. Pn/Po has been included in the figure caption.
- Page 7, Lines 204-207. The observed effect is mainly due to the presence of pectinolytic enzymes, right?
R: The sentence has been rephrased highlighting the effect of protease addition at dosage higher than 10 μL/L in lowering the protein instability. However, it is worthy to describe the effect of pectinolytic enzyme alone (Ctrl) in lowering the protein instability as compared to the turbid pressed grape juice (cold-settled without any enzyme addition).
- Figure 3. What exactly is ΔNTU and how has it been calculated for each sample in both treatments? It does not seem that the reference is the same and, therefore, the results would not be comparable.
R: In both treatments ΔNTU has been calculated as described in section 2.4. Now the reference is Ctrl in both treatments (I and II).
- Ctrl is defined differently in the different figures. Is each case a different sample?
R: Sorry for the mistake. Ctrl has been uniformly defined in the different figures.
- The figure 4a confirms the idea that the observed effect is mainly due to the presence of pectolytic enzymes and these are not present in all the samples, so the conclusions obtained in this work should be questioned.
R: As stated above, the pectinolytic preparation was applied in both treatments and the only variable is the amount of microbial protease added. Thus, the effect observed may be uniquely ascribable to the presence of microbial protease at different dosage and the conclusion should be not questioned.
- Page 11, Lines 272-274. Two different effects are then compared since the two treatments are not carried out under the same conditions in relation to the presence/absence of pectolytic enzymes. Two variables should be considered for the study to be complete and the results obtained objective.
R: We would like to thank the Reviewer for her/his observation. Data in Table 4 have been revised recalculating ΔE* and always referring the values to Ctrl sample, as now indicated in section 2.6.
- Figure 7a: Indicators of significant differences do not appear
R: We apologize for the omission. In Figure 7a indicators of significant differences have been added.
Reviewer 2 Report
The objective of the work was to study the use of a commercial food-grade protease from Aspergillus niger in the prevention of protein haze in white wines. The focus of the study is very interesting and important for winemakers, technicians, and researchers in oenology.
The paper is very well written. The introduction describes exhaustively and concisely the state of the art and the different aspects involved in the subject. The different essays and analysis procedures are clearly described. The results were analyzed in detail and the discussion is successful. The conclusions are very well written, as they highlight the main results and applications that can be made from this work.
The authors use a ‟wine-must″ model to study the enzymatic activity. This model has the normal pH of musts and wines, but this ‟wine-must″ does not contain ethanol. Could the authors briefly explain why they used a model without ethanol and how they think this compound (formed during winemaking and therefore in a variable content during Treatment II) would affect the activity of the proteases?
On the other hand, when referring to the temperatures of the essays, the authors indicate that a ‟low temperature″ (10 °C) and a ‟medium temperature″ (18 °C) are employed. I think that these definitions may be unclear, since 18 °C is a normal temperature to produce white wines. Perhaps ‟medium temperature″ could be replaced by normal or usual temperature, or simply clarify that this temperature was chosen because it is the one that is usually used in white wine fermentation.
The first paragraphs of point 3.3 (lines 197 to 203) are repetitive and could be removed from the text, or at least summarized.
When referring to the CIELab parameters, the authors alternatively indicate hue (h*) and Hue (H*) (lines 267 and 278). I think it would be best including the L* values to better describe the color.
On the other hand, when referring to the temperatures of the essays, the authors indicate that a ‟low temperature″ (10 °C) and a ‟medium temperature″ (18 °C) are employed. I think that these definitions may be unclear, since 18 °C is a normal temperature to produce white wines. Perhaps ‟medium temperature″ could be replaced by normal or usual temperature, or simply clarify that this temperature was chosen because it is the one that is usually used in white wine fermentation.
The first paragraphs of point 3.3 (lines 197 to 203) are repetitive and could be removed from the text, or at least summarized.
When referring to the CIELab parameters, the authors alternatively indicate hue (h*) and Hue (H*) (lines 267 and 278). I think it would be best including the L* values to better describe the color.
Author Response
The objective of the work was to study the use of a commercial food-grade protease from Aspergillus niger in the prevention of protein haze in white wines. The focus of the study is very interesting and important for winemakers, technicians, and researchers in oenology.
The paper is very well written. The introduction describes exhaustively and concisely the state of the art and the different aspects involved in the subject. The different essays and analysis procedures are clearly described. The results were analyzed in detail and the discussion is successful. The conclusions are very well written, as they highlight the main results and applications that can be made from this work.
- The authors use a ‟wine-must″ model to study the enzymatic activity. This model has the normal pH of musts and wines, but this ‟wine-must″ does not contain ethanol. Could the authors briefly explain why they used a model without ethanol and how they think this compound (formed during winemaking and therefore in a variable content during Treatment II) would affect the activity of the proteases?
R: An excellent point was mentioned by the reviewer. Accordingly to previous studies, performed by our research group, ethanol at medium content in wine (13% v/v) slightly affects the activity of different enzymes, including proteases.
- On the other hand, when referring to the temperatures of the essays, the authors indicate that a ‟low temperature″ (10 °C) and a ‟medium temperature″ (18 °C) are employed. I think that these definitions may be unclear, since 18 °C is a normal temperature to produce white wines. Perhaps ‟medium temperature″ could be replaced by normal or usual temperature, or simply clarify that this temperature was chosen because it is the one that is usually used in white wine fermentation.
R: We are grateful to Reviewer for her/his suggestion. In section 2.3 we have indicated that 18 °C is the temperature usually used in white wine fermentation.
- The first paragraphs of point 3.3 (lines 197 to 203) are repetitive and could be removed from the text, or at least summarized.
R: According to Reviewer’s suggestion the first paragraphs of point 3.3 has been summarized.
- When referring to the CIELab parameters, the authors alternatively indicate hue (h*) and Hue (H*) (lines 267 and 278). I think it would be best including the L* values to better describe the color.
R: hue (h*) has been fixed and L* values have been added.
Round 2
Reviewer 1 Report
Reference 34 should be revised.